# Gene-Activated Matrix with Self-Assembly Anionic Nano-Device Containing Plasmid DNAs for Rat Cranial Bone Augmentation

**DOI:** 10.3390/ma14227097

**Published:** 2021-11-22

**Authors:** Masahito Hara, Yoshinori Sumita, Yukinobu Kodama, Mayumi Iwatake, Hideyuki Yamamoto, Rena Shido, Shun Narahara, Takunori Ogaeri, Hitoshi Sasaki, Izumi Asahina

**Affiliations:** 1Department of Regenerative Oral Surgery, Unit of Translational Medicine, Nagasaki University Graduate School of Biomedical Science, 1-7-1 Sakamoto, Nagasaki 852-8588, Japan; bb55317205@ms.nagasaki-u.ac.jp (M.H.); yamamotohideyuki@nagasaki-u.ac.jp (H.Y.); r-shido@nagasaki-u.ac.jp (R.S.); narashun@nagasaki-u.ac.jp (S.N.); asahina@nagasaki-u.ac.jp (I.A.); 2Basic & Translational Research Center for Hard Tissue Disease, Nagasaki University Graduate School of Biomedical Sciences, 1-7-1 Sakamoto, Nagasaki 852-8588, Japan; iwatake@nagasaki-u.ac.jp (M.I.); ogaeri@nagasaki-u.ac.jp (T.O.); 3Department of Hospital Pharmacy, Nagasaki University Hospital, 1-7-1 Sakamoto, Nagasaki 852-8501, Japan; y-kodama@nagasaki-u.ac.jp (Y.K.); sasaki@nagasaki-u.ac.jp (H.S.)

**Keywords:** bone augmentation, in vivo gene delivery, gene-activated matrix, plasmid vector, self-assembly nano-device

## Abstract

We have developed nanoballs, a biocompatible self-assembly nano-vector based on electrostatic interactions that arrange anionic macromolecules to polymeric nanomaterials to create nucleic acid carriers. Nanoballs exhibit low cytotoxicity and high transfection efficiently in vivo. This study investigated whether a gene-activated matrix (GAM) composed of nanoballs containing plasmid (p) DNAs encoding bone morphogenetic protein 4 (pBMP4) could promote bone augmentation with a small amount of DNA compared to that composed of naked pDNAs. We prepared nanoballs (BMP4-nanoballs) constructed with pBMP4 and dendrigraft poly-L-lysine (DGL, a cationic polymer) coated by γ-polyglutamic acid (γ-PGA; an anionic polymer), and determined their biological functions in vitro and in vivo. Next, GAMs were manufactured by mixing nanoballs with 2% atelocollagen and β-tricalcium phosphate (β-TCP) granules and lyophilizing them for bone augmentation. The GAMs were then transplanted to rat cranial bone surfaces under the periosteum. From the initial stage, infiltrated macrophages and mesenchymal progenitor cells took up the nanoballs, and their anti-inflammatory and osteoblastic differentiations were promoted over time. Subsequently, bone augmentation was clearly recognized for up to 8 weeks in transplanted GAMs containing BMP4-nanoballs. Notably, only 1 μg of BMP4-nanoballs induced a sufficient volume of new bone, while 1000 μg of naked pDNAs were required to induce the same level of bone augmentation. These data suggest that applying this anionic vector to the appropriate matrices can facilitate GAM-based bone engineering.

## 1. Introduction

At present, autogenous bone grafts, also known as autografts, are still widely considered to be the “gold standard” augmentation technique for facial and orbital reconstruction, especially for the purpose of regenerating bony defects in the craniofacial region [1,2]. However, autografts are sometimes associated with donor-site morbidity and a lack of tissue availability [1,2,3]. As a result, bone engineering, which is considered to be safer and less invasive and to have higher efficacy than conventional treatments such as autogenous bone grafts, is increasingly being considered as a bone reconstruction strategy. Consequently, increasing interest has also been seen in morphogens and growth factors as key elements in conferring osteoinducibility to artificial bone substitutes in bone engineering [4,5,6]. However, there are some difficulties in controlling their bioactivities at the transplanted site due to several disadvantages, such as early inactivation with short half-lives and potential side effects [7,8]. Among such morphogens and growth factors, bone morphogenetic proteins (BMPs), such as recombinant (r) BMP2 or rBMP4, have been reported to induce bone formation in various settings [4,9,10]. However, high doses of rBMP2 have also been reported to induce substantial swelling, which can lead to airway obstruction, when implanted directly to oral and cervical areas [11]. Therefore, to advance bone engineering, the development of a more efficient and controlled delivery system is needed [6].

The gene-activated matrix (GAM) enables safer gene delivery compared to protein delivery and thus has potential usefulness as an alternative method for protein delivery in bone engineering [12]. GAM, which consists of gene vectors that encode target proteins and a biodegradable matrix such as collagen, is able to release gene vectors to the surrounding tissues slowly and is able to express proteins at physiological concentrations long after transplantation [12,13,14]. Therefore, the development of effective conditions for transferring genes in vivo without any viral vectors or cytotoxic transfection materials/reagents is needed to facilitate the use of this strategy in the clinical setting [15,16]. Efficient gene transfer is possible through viral vectors, but this involves a number of disadvantages, including immunogenicity, a risk of virus-dependent recombination, and protein expression exceeding that required for tissue recovery [16,17]. Therefore, nonviral plasmid vectors have frequently been adopted to help induce bone regeneration, despite unresolved problems such as a low efficiency of transfection [11,16,18]. For this reason, we previously developed and subsequently demonstrated the potential of an atelocollagen-based GAM that contains naked plasmid (p) DNAs encoding osteogenic proteins such as pBMP4 that is safe for simple bone engineering [19]. However, even after incorporating pBMP4, a substantial dose (>6 μg/μL) of pDNA was needed to induce a sufficient volume of newly formed bone in rat crania. Because of that, to improve the transgene efficacy, we have recently investigated the applying microRNA (miR) 20a to an atelocollagen-based GAM [20]. miR20a can simultaneously regulate multiple pathways including BMP/Runx2 and angiogenic signaling, each of which is essential to successful bone regeneration/augmentation. [6,20]. This study suggested that miR delivery strongly promotes the osteoinducibility of GAM and reduces the amount of its expression plasmids, but this study still required 3 µg/µL of pmiR20a. Therefore, the further development of gene delivery devices that display the low toxicity and high transfection efficacy of genes to osteogenic cells is extremely crucial in GAM-based bone engineering.

Recently, we developed a self-assembly nano-device, which we called nanoballs, that contains pDNAs for in vivo gene transfection [21]. Nanoballs are composed of cationic cores, which are constructed with pDNA and cationic polymers such as dendrigraft poly-L-lysine (DGL) and are coated with anionic polymers such as γ-polyglutamic acid (γ-PGA). In general, among nonviral vectors, cationic polymers are frequently employed to form stable cationic complexes with pDNA and display high transfection efficacy in vivo [22]; however, cationic complexes are cytotoxic because of direct interactions with negative charges on cell surfaces [23,24]. Therefore, coating cationic complexes with anionic polymers can decrease their cytotoxicity without reducing their efficacy [24]. Typically, because they repulse the cellular membrane electrostatically, anionic complexes are not taken up well by cells. However, by screening various anionic polymers, we discovered that a γ-PGA-coated vector has low cytotoxicity and high transfection efficiency that is comparable to that of cationic complexes because γ-PGA binds with a specific receptor on the cell surface [24]. Indeed, several anionic polymers have been shown to be taken up by macrophages or endothelial cells through specific receptors such as scavenger and Toll-like receptors [25,26]. We confirmed that transgene expression efficiency was the highest in the cells of the splenic marginal zone when administering γ-PGA-coated nanoballs intravenously into mice (40 μg DNA per mouse) [21]. In addition, such DNA nanoballs can transfect genes efficiently into antigen-presenting cells and can serve as a vaccine vector for the suppression of malaria or melanoma [27,28]. Moreover, nanoballs should be safe since they are constructed with safe materials that are already used in medicines and foods, such as DGL and γ-PGA, and can be prepared as a sterilized product at a large scale [21,29]. Therefore, nanoballs can provide higher efficacy for in vivo gene transfer with GAM.

In this study, we investigated whether atelocollagen-based GAM containing nanoball vectors encoding pBMP4 (BMP4-nanoballs) could facilitate rat cranial bone augmentation, which is a definitive model of regenerative therapy for jawbone atrophy. As severe jawbone atrophy continues to be a major problem in the oral and maxillofacial area, the development of new osteoinductive biomaterials is urgently needed. As mentioned above, we previously confirmed that atelocollagen-based GAM can reliably induce engineered bone in rats when naked pBMP4 or pmiR20a are incorporated at a high dose (3–6 μg/μL: 500–1000 μg of pDNA in 7 mm diameter × 2 mm thickness of columnlike GAM) [18,20]. In addition, cationic complexes such as polyethyleneimine (PEI) with pBMP2 have shown promise in gene delivery for rat calvaria bone engineering at low doses (e.g., 10–25 μg) of pDNA or by chemically modified RNA (cmRNA) in 5 mm diameter × 2 mm thickness scaffolds [30,31]. Therefore, we hypothesized that GAM composed of anionic nanoballs may be effective for bone augmentation using less pDNA without the cytotoxicity caused by cationic complexes because of direct interactions with negative charges on cells.

## 2. Materials and Methods

### 2.1. Plasmid Preparation

All procedures involving animals in this study were conducted in accordance with the relevant ethical guidelines, and the study protocol and procedures were approved by the Nagasaki University Animal Ethics and Use of Committee (1812051492, Nagasaki, Japan). Standard recombinant PCR methods and confirmed nucleotide sequencing were used to construct all of the expression vectors. Mouse (m) bmp4 cDNA was obtained from mouse calvaria at embryonic day 18.5. Total RNA was extracted using TRIzol (Invitrogen, Carlsbad, CA, USA), and reverse transcription was performed using the ReverTra Ace^®^ qPCR RT Master Mix with gDNA Remover according to the manufacturer’s instructions (Toyobo, Osaka, Japan), with specific primer sets constructed using the National Center for Biotechnology Information reference sequence (BMP4; NM_007554.2). Forward and reverse primers involved the Kozak sequence and the XhoI site and included the SalI site for ligation (bmp4 primer pair: forward 5′-ctcgaggccaccatgattcctggtaaccgaatgc-3′ and reverse 5′-GTCGACTCAGCGGCATCCACACCC-3′), respectively. The pAcGFP-N1 vector (Clontech, Palo Alto, CA, USA) linked to an internal ribosomal entry site (pIRES-AcGFP; kindly supplied by Dr. T. Komori, Nagasaki University, Nagasaki, Japan) was prepared using pDNA encoding green fluorescent protein (pGFP) as a control. Next, vectors and cDNAs were ligated (pBMP4) following the digestion of the pIRES-AcGFP vector and cDNAs using XhoI and SalI. All of the newly generated expression plasmids in this study were verified using the BigDye^®^ Terminator v3.1 cycle Sequencing Kit (Invitrogen).

### 2.2. Preparation of Nanoballs

The 5G DGL compounds (MW: 172,300 Da, 963 lysine groups) were purchased from COLCOM S.A.S. (Montpellier, France). Yakult Pharmaceutical Industry Co., Ltd. (Tokyo, Japan) provided the γ-PGA. The resultant pDNAs (pGFP or pBMP4) were then dissolved in 5% dextrose solution and stored at −80 °C until analysis. The pDNA concentration of the solution was calculated by measuring the absorbance of the solution at 260 nm and then adjusted to 1 mg/mL. Next, the pDNA and DGL solutions (pH 7.4) were mixed thorough pipetting and were then left at room temperature for 20 min to prepare the binary complexes. As a result, pDNA/DGL complexes with charge ratios of 1:6 (DGL 6 complexes) were produced (Figure 1A). Finally, γ-PGA was mixed with the DGL 6 complexes by pipetting them in order to produce complexes with charge ratios of 1:6:8 (DGL 6/PGA 8 complexes); these were left at room temperature for another 20 min to construct the ternary complexes (GFP-nanoballs or BMP4-nanoballs) (Figure 1A).

The morphology of the complexes was observed using a transmission electron microscope (JEM-1230; JEOL Ltd., Tokyo, Japan) on 80 kV acceleration voltage and was pictured using a 2 k × 2 k Veleta CCD camera (Olympus Soft Imaging Solutions, Lakewood, CO, USA) (Figure 1B). A Zetasizer Nano ZS (Malvern Instruments, Ltd., Malvern, UK) was used to measure the ζ-potentials and particle sizes, shown as the number-weighted mean diameter, of each complex (Figure 1C). Next, 20-μL aliquots of each complex solution containing 1 μg pDNA were mixed with 4 μL loading buffer (30% glycerol and 0.2% bromophenol blue) and were loaded onto 0.8% agarose gels to assess the complex formation. Electrophoresis using the i-Mupid J system (Cosmo Bio, Tokyo, Japan) was carried out for 60 min at 50 V in running buffer solution (40 mM Tris/HCl, 40 mM acetic acid, and 1 mM ethylenediaminetetraacetic acid (EDTA)), and ethidium bromide staining was used to visualize the pDNA retardation.

### 2.3. Cell Culture

For the isolation of the macrophages from rat bone marrow, muscles were removed to expose the femur and tibia after sacrificing Wistar rats (male, 6–7 weeks old; CLEA Japan Inc., Tokyo, Japan). The bone marrow was flushed out using Dulbecco’s modified Eagle medium (DMEM; Sigma Aldrich, St. Louis, MO, USA). Then, the medium was collected. After centrifugation (1800 rpm, 5 min), lysis buffer was applied for 5 min and EDTA for 10 min to remove the red blood cells. After washing with phosphate-buffered saline (PBS), the medium containing bone marrow cells was filtered through a 70-μm cell strainer (Corning Falcon, Corning, NY, USA). Then, the cells were cultured with DMEM containing 10% fetal bovine serum (FBS), 2% antibiotic antimycotic solution, and Mirimostim (Leukoprol; JCR Pharm, Inc., Hyogo, Japan) in 10 cm plates at 37 °C under 5% CO_2_. On the next day of culture, the medium was changed, and the macrophages were removed to new 10 cm plates containing Leukoprol and medium. The cells were cultured as macrophages for the subsequent experiments.

For the isolation of mesenchymal stem/progenitor cells (MSCs) from rat compact bone, muscles were removed to expose the femur and tibia after sacrificing Wistar rats (male, 6–7 weeks old). Then, the bones were crushed into small pieces using bone scissors. After washing with PBS, the bone fragments were digested using 1 mg/mL collagenase II (Sigma Aldrich) in DMEM containing 10% FBS (Sigma Aldrich) and 2% antibiotic antimycotic solution on a shaker for 1 h at 37 °C. The digests were filtered through a 70-μm cell strainer, and the bone fragments remaining on the cell strainer were washed three times with PBS. The bone fragments were cultured with DMEM containing 10% FBS and 2% antibiotic antimycotic solution in 10 cm plates at 37 °C under 5% CO_2_. On the third day of culture, the medium was changed, and the bone fragments were removed. When the cells reached 80% confluence, they were subcultured as MSCs until passage three (P3) for subsequent experiments.

For the isolation of the fibroblasts from the rat auricles, epithelium was removed by using 1 mg/mL Dispase II (Godo Shusei Co., Ltd., Tokyo, Japan) in DMEM containing 10% FBS and 2% antibiotic antimycotic solution on a shaker for 1 h at 37 °C to expose the dermis after sacrificing Wistar rats (male, 6–7 weeks old). Then, the dermis was cut into small pieces using scissors. After washing with PBS, the dermis fragments were cultured with DMEM containing 10% FBS and 2% antibiotic-antimycotic solution in 10 cm plates at 37 °C under 5% CO_2_. At 1 week of culture, medium was added to prevent drying, and the dermis fragments were removed. When the remaining cells reached 80% confluence, they were subcultured as fibroblasts until P3 for subsequent experiments.

### 2.4. Functional Analysis of BMP4-Nanoballs In Vitro

A total of 5 × 10^5^ macrophages, MSCs, or fibroblasts were seeded on 10 cm plates. Then, after 24 h, culture medium (DMEM containing 10% FBS and 2% antibiotic-antimycotic solution) was replaced with medium mixed with 5 μg of pGFP- or pBMP4-nanoball vectors. At 2 h post-transfection, the transfection medium was removed and replaced with the original medium, and cells were subsequently cultured for 48 h. Then, at 48 h post-transfection, the transfected cells in the cultured macrophages, MSCs, and fibroblasts were observed for GFP expression under a confocal microscope (LSM 800 with Airyscan; Carl Zeiss, Inc., Oberkochen, Germany). Then, total RNAs were extracted from the cultured cells by using TRI Reagent. Next, qPCR was used to detect the expressions of *bmp4* mRNA in the specimens. The ReverTra Ace^®^ qPCR RT Kit with gDNA Remover (Toyobo) was used for the cDNA synthesis. Then, qPCR was performed using SYBR green and gene-specific primers on an Mx3000p real-time PCR system. Table 1 shows the rat-specific primer sets; glyceraldehyde-3-phosphate dehydrogenase (*gapdh*) was used as the internal standard.

After transfection, macrophages, MSCs, and fibroblasts were fixed with 4% paraformaldehyde, washed three times with PBS, and treated with 100 mM glycine buffer. Then, the cells were washed again three times and incubated at 4 °C overnight with rabbit anti-BMP4 antibody (1:200) (Table 2). Next, the slides were incubated for 2 h at room temperature with goat anti-rabbit secondary antibody (1:200) (Table 2). After washing three times with PBS, the cells were counterstained with Vectashield mounting medium containing 4′,6′-diamino-2-phenylindole (Vector Laboratories Inc., Burlingame, CA, USA) and were observed using a confocal microscope (LSM 800 with Airyscan; Carl Zeiss, Inc.).

### 2.5. Functional Analysis of BMP4-Nanoballs In Vivo

To assess the biological function of the BMP4-nanoballs in vivo, 5 μg of GFP- or BMP4-nanoballs in 20 μL sterile water were seeded to an Octacalcium phosphate/Collagen (OCP/Col) disk (Toyobo, Otsu, Japan) 4 mm in diameter and 1.5 mm in thickness. Next, the nanoballs were immediately transplanted into the rat (11-week-old male Wistar rats) calvaria bone defects (4 mm in diameter) (three rats per group) after the rats had been anesthetized through the inhalation of isoflurane (introduction concentration: 4–5%, maintenance concentration: 2–3%; FUJIFILM Wako Pure Chemical Corp., Osaka, Japan). Then, the specimens were harvested at 4 weeks of transplantation, and the acceleration of bone reconstruction was analyzed using a micro-computed tomography (micro-CT) system (R_mCT2; Rigaku Corp., Tokyo, Japan) with a voxel resolution of 20 μm and an energy level of 90 kV. The three-dimensional (3D) micro-CT images were then reconstructed and morphometrically analyzed using TRI/3D-BON structural analysis software (Ratoc System Engineering, Tokyo, Japan).

### 2.6. Manufacture of GAM Containing Nanoballs

The GAMs used in the experiments in this study were all prepared the day before transplantation. Amounts of 1, 5, 10, 25, 50, or 100 μg of BMP4-nanoballs or 1000 μg of naked-AcGFP plasmid vector-harboring BMP4 cDNA in 60 μL of sterile water were mixed well with 20 mg β-tricalcium phosphate (β-TCP) granules (0.5–1.5 mm size; Osferion, Olympus, Tokyo, Japan) and 100 μL of 3% bovine atelocollagen (Atelocollagen Implant; Koken, Tokyo, Japan) inside the lids of 1.5-mL Eppendorf tubes; these mixtures were then lyophilized overnight. The GFP-nanoballs and vehicle AcGFP plasmids alone were used as the experimental controls. The experimental groups included those transfected with GFP-nanoballs, naked pGFP (as a control), BMP4-nanoballs, and naked pBMP4.

A manufactured GAM with 5 μg of BMP4-nanoballs was vertically or horizontally split, and the surface was coated with gold ions (IB-2; EIKO Engineering, Tokyo, Japan). The 3D structure of GAM was observed under a scanning electron microscope (SEM) (Miniscope^®^ TM-1000; HITACHI High-Tech Corp., Tokyo, Japan). The image magnification ranged from 100× to 10,000×. In this study, the accelerating voltage was set to 15 kV, and the contrast and brightness of the images were automatically adjusted to optimal values.

### 2.7. GAM Transplantation

Six- to seven-week-old male Wistar rats were anesthetized through the inhalation of isoflurane (introduction concentration: 4%–5%, maintenance concentration: 2–3%; FUJIFILM Wako Pure Chemical Corp.). All rats were kept warm during and after surgery. GAMs with GFP- or BMP4-nanoballs were then transplanted to the cranial bone surface under the periosteum (*n* = 240; 5 rats/group, with each group containing 1, 5, 10, 25, 50, and 100 μg of pGFP- or pBMP4-nanoballs, respectively) at each point in time (1-, 2-, 4-, and 8-weeks post-transplantation) as a model of alveolar bone augmentation (Appendix A). Furthermore, GAMs with naked pGFP or pBMP4 were also examined for transplantation (*n* = 20; 5 rats/group, with each group containing 1000 μg naked pGFP or pBMP4, respectively) at each point in time (4- and 8-weeks post-transplantation). At 1- and 2-weeks post-transplantation, the specimens were harvested to detect the transfected cells and to determine the specific gene expressions. At 4- and 8-weeks post-transplantation, the specimens were analyzed using micro-CT (R_mCT; Rigaku Corp.) and were harvested to evaluate the transfection efficiency and to examine new bone formation histologically.

### 2.8. Characteristics of Transfected Cells: Flow Cytometry Analysis

At 1- and 2-weeks post-transplantation, the transplanted GAMs were harvested to detect transfected cells. Harvested specimens were cut into small pieces using scissors. After washing with PBS, the minced tissues were digested with 1 mg/mL collagenase II in DMEM on a shaker for 1 h at 37 °C. Then, the digested cells were isolated by filtering through a 70-μm cell strainer. Next, to detect the surface antigen positivity of mesenchymal (CD90^+^) or macrophage (CD11b^+^) subpopulations as well as GFP-positive cells, the cells that had been freshly isolated from the specimens were subjected to flow cytometry; Table 2 lists the antibodies that were used. The cells were then suspended in 2 mmol/l of EDTA/0.2% bovine serum albumin (BSA)/PBS buffer (5 × 10^5^ cells/200 μL), incubated at 4 °C for 30 min following the addition of 10 μL of FcR blocking reagent, and then dispensed equally into reaction tubes (100 μL/tube) for staining. Next, each aliquot was incubated at 4 °C for 20 min with 2 μL of each 1st-Ab and washed twice with 1 mL of 2 mmol/L EDTA/0.2% BSA/PBS buffer. The cells were then resuspended in 2 mmol/L of EDTA/0.2% BSA/PBS buffer (2 × 10^5^ cells/200 μL), and an LSRFortessa cell analyzer (BD Biosciences, San Jose, CA, USA) with FlowJo software (Tomy Digital Biology Co., Ltd., Tokyo, Japan) was used to perform flow cytometry analysis. Finally, the ratio of CD90- or CD11b-positive subpopulations in the GFP-positive cells per each gate among those isolated from specimens at 1- or 2-weeks post-transplantation was assessed and calculated in relation to that of the whole cell population.

### 2.9. Characteristics of Transplants: Expressions of Osteoblast- and Macrophage-Related Genes

At 1- and 2-weeks post-transplantation, the transplanted GAMs were harvested and pulverized using a homogenizer (MP-Biomedicals, Tokyo, Japan). Total RNAs were extracted using TRI Reagent. Next, qPCR was employed to detect the mRNA expressions of the osteogenic (*bmp4* and *osteocalcin*) and macrophage (*f4/80* and *cd206*) genes in the specimens. The ReverTra Ace^®^ qPCR RT Kit with gDNA Remover (Toyobo) was used for cDNA synthesis. Then, qPCR was performed using SYBR green and gene-specific primers on an Mx3000p real-time PCR system. Table 1 shows the rat-specific primer sets; glyceraldehyde-3-phosphate dehydrogenase (*gapdh*) was employed as the internal standard.

### 2.10. Histological Observations

Next, the specimens were harvested and fixed with 4% paraformaldehyde to assess bone augmentation at 4- and 8-weeks post-transplantation. The specimens were then decalcified using a solution containing 2.9% citric acid, 1.8% trisodium citrate dehydrate, 10% formic acid, and 90% distilled water and were embedded in paraffin. Deparaffinized sections (5-μm thick) were then stained with hematoxylin and eosin (H&E). NIH ImageJ software (NIH, Bethesda, MD, USA) was used to analyze the volume of augmented bone-like tissue, and the percentage of surface area occupied by bone-like tissue was observed under light microscopy (30× magnification) using five sections from each specimen per group (five specimens in a group). Sections were chosen randomly and independently by two examiners, and the new bone areas were measured by pixels.

Next, immunohistochemical staining was performed on the specimens at 8-weeks post-transplantation using a Vectastain ABC kit (Vector Laboratories Inc.). As shown in Table 2, the sections were stained with rabbit polyclonal anti-osteocalcin (1:200) for osteogenic cells and new bone tissues and with rabbit polyclonal anti-GFP antibodies (1:500) for transfected cells, and then the slides were incubated with anti-rabbit secondary antibody (1:300). Next, the specimens were reacted with 0.1% *w/v* 3.3′-diaminobenzidine tetrahydrochloride (DAB; GenWay Biotech, San Diego, CA, USA) in PBS were and counterstained with hematoxylin. Control staining was conducted by replacing the first antibody with pre-immune serum eluted from the corresponding affinity columns. In total, five sections were stained from each of the five specimens per group.

### 2.11. Statistical Analysis

One-way analysis of variance was conducted to analyze the means, and Dunnett’s multiple comparison *t*-test was used to identify significant differences within each group. Experimental values are expressed as mean ± standard deviation (SD). *p*-values < 0.05 were considered statistically significant.

## 3. Results

### 3.1. Biological Activity of BMP4-Nanoballs In Vitro and In Vivo

To confirm the biological activity of BMP4-nanoballs, we first analyzed whether the nanoballs could effectively transfer the genes into cultured macrophages, MSCs, and fibroblasts, which migrate or infiltrate to bone regeneration sites (Figure 2A), and we observed BMP4 production from these cells. As a result, at 2 h post-transfection, GFP signals were detectable in each cell type (Figure 2B). At 48 h after transfection, BMP4 mRNA expression was significantly upregulated in macrophages, MSCs, and fibroblasts via transfection with BMP4-nanoballs compared with that in cells transfected with GFP-nanoballs (Figure 2C). Consistent with this result, BMP4 production was recognized in many cells after treated with BMP4-nanoballs when each cell type was stained with an anti-BMP4 antibody (Figure 2D). In contrast, there were few stained cells after the transfection of the GFP-nanoballs in each cell type (data not shown).

To evaluate the biological function of the BMP4-nanoballs in vivo, the nanoballs were delivered by OCP/Col bone substitutes into rat calvaria bone defects (Figure 3A,B). At 4 weeks after post-transplantation, micro-CT analyses revealed that the BMP4-nanoballs accelerated bone formation ubiquitously at the transplanted sites compared to the GFP-nanoballs (Figure 3C). Indeed, when the bone volume (BV), bone mineral content (BMC), bone mineral density (BMD), and bone volume/tissue volume (BV/TV) were analyzed, the BMP4-nanoballs showed a significant increase (approximately twofold) in BV, BMC, and BV/TV values compared to the GFP-nanoballs (Figure 3D).

### 3.2. Manufactured GAM for Bone Augmentation

To deliver BMP4-nanoballs for bone augmentation, we employed the atelocollagen gel and β-TCP granules as delivery carriers and bone substitutes. At the transplanted sites, atelocollagen favors the sustained release of BMP4-nanoballs, and the β-TCP granules can maintain the augmented space (Figure 4A). The gross appearance of the manufactured GAM was a cylindrical sponge-like shape (8 mm in diameter and 3 mm in thickness) (Figure 4B). The morphology of GAM before transplantation was examined by an SEM (Figure 4C–G). At low magnification, the flat form of the porous atelocollagen sponge and β-TCP granules were observed (Figure 4C,D), and many micro- or nano-sized β-TCP particles were seen at the surface of the atelocollagen sponge (Figure 4E). Some of the uniform nano-sized particles that seemed to be nanoballs were detectable on the flat form of the atelocollagen sponge at a higher magnification (Figure 4F,G).

### 3.3. Detection of Transfected Cells and Gene Expression Related to New Bone Formation in Transplanted GAMs

After transplanting GAMs onto the rat calvaria bone under the periosteum (Figure 5A), the transfected cells in those specimens were investigated for whether they were present as GFP-expressing cells in GAMs with or without BMP4-nanoballs. At 1- and 2-weeks post-transplantation, the percentages of GFP-expressing cells in CD90-positive mesenchymal progenitors and CD11b-positive macrophages isolated from GAM specimens were analyzed. The abundance of transfected cells (CD90+/GFP+) among the mesenchymal progenitor cells was approximately 0.78 ± 0.049% at 1 week but increased to 1.86 ± 0.105% at 2 weeks (Figure 5B,C). Meanwhile, the percentages of transfected (CD11b+/GFP+) macrophages were approximately 1.39 ± 0.234% and 1.65 ± 0.072% at 1 and 2 weeks, respectively (Figure 5B,D). This result indicates that the macrophages that were initially infiltrating into the GAM took up nanoballs well, and then some of the migrated mesenchymal progenitor cells took up nanoballs and were transfected.

Moreover, for osteoblast and macrophage differentiation markers, higher expressions of BMP4 and CD206 mRNAs were recognized in GAM specimens with BMP4-nanoballs at 1 and 2 weeks compared to those in specimens with GFP-nanoballs (Figure 6A,B). In addition, while maintaining higher expressions of BMP4 and M2-macrophage (F4/80 and CD206) genes, the increased expression of osteocalcin mRNA was clearly found in specimens with BMP4-nanoballs at 2 weeks (Figure 6A). These results suggest that osteoblastic and anti-inflammatory macrophage differentiations were promoted from the initial stage by transfection with pBMP4 and that the osteoblasts then increased over time in GAM specimens with BMP4-nanoballs.

### 3.4. Histological Analysis of Bone Augmentation

For 4 to 8 weeks following the transplantation of GAM with 5 μg of GFP- or BMP4-nanoballs, the promotion of bone formation was recognized on micro-CT analysis (Figure 7A,B). We transplanted GAMs with various amounts (1, 5, 10, 25, 50, and 100 μg) of GFP- or BMP4-nanoballs into the rats. Promoted bone formation was observed in GAM specimens with 1–10 μg of BMP4-nanoballs at 4 weeks compared to specimens with the same amounts of GFP-nanoballs (Figure 7C–F). In those specimens, new replacement bone tissue with osteocytes was recognized at the surfaces of absorbed β-TCP granules (Figure 7H). At 8 weeks after post-transplantation, we found that the bone tissues were largely augmented when transplanted with GAMs with 1–10 μg of BMP4-nanoballs (Figure 7D–F), whereas no obvious new bone formation was detectable in GAMs with GFP-nanoballs (Figure 7C). The augmented bone seemed to be mature in the GAMs specimens with 1–10 μg BMP4-nanoballs because new replacement bone tissue stained by anti-osteocalcin antibody was recognized at distant sites from the host bone (Figure 7I,J). Furthermore, a few transfected cells (as GFP-expressing cells) were also observed at the surface of β-TCP granules at distant sites from the host bone (Figure 7I). However, sufficient volumes of new bone formation were not found when the transplanted GAMs contained more than 25 μg of BMP4-nanoballs because of the aggregation of broken nanoballs in the GAMs (data not shown).

The amounts of newly formed bone by GAMs harboring 1, 5, or 10 μg of BMP4-nanoballs at 4 weeks were 14,615.6 ± 1938.9, 17,450.8 ± 3092.6, and 14,913.1 ± 2376.0 pixels, respectively, in the entire transplant area. Those amounts were comparable to that generated by 1000 μg of naked pBMP4 (13,815.6 ± 1865.8 pixels), whereas that by GAMs with GFP-nanoballs was 5842.5 ± 528.5 (Figure 8). Subsequently, the amounts of augmented bone by GAMs harboring 1, 5, or 10 μg of BMP4-nanoballs at 8 weeks reached 20,818.8 ± 2913.0, 24,697.6 ± 5164.9, and 20,244.5 ± 4495.0 pixels, respectively. Those amounts were also comparable to that by 1000 μg of naked pBMP4 (20,971.2 ± 5823.4 pixels); however, the peak achieved by the GAMs with the GFP-nanoballs only reached 6401.7 ± 1025.8 pixels (Figure 8).

## 4. Discussion

In this study, we validated in vivo gene delivery by the anionic nanoball vector with alloplastic substitutes, which may provide a straightforward and effective strategy for bone augmentation. The positive outcomes of this study were as follows: (1) after in vitro transfection, nanoball vectors containing pBMP4 reliably induced BMP4 protein production in cells that would migrate to the bone formation site; (2) this vector was also gradually taken up by cells that infiltrated into the transplanted GAMs, leading to the promotion of osteogenic gene expression over time; and (3) the resultant bone augmentation was eventually promoted when 1–10 μg of BMP4-nanoballs was incorporated into atelocollagen-based GAMs. These outcomes suggest that the ternary complex of pDNA, DGL, and γ-PGA is particularly useful for developing clinically applicable GAMs.

Regarding GAM strategies using nonviral vectors, synthetic or natural polymers have been investigated widely as vector materials; as a result, several advantages in vivo for designing a vehicle with well-defined structural and chemical properties have been reported, including non-immunogenicity, low acute toxicity, and flexibility [16,17,24]. These polymers are usually cationic and simply complexed to pDNA with electrostatic interactions. Among these cationic polymers, PEI is a popular synthetic material that can be used to form complexes and can bind to the cell surface, be taken up by the endocytic pathway, or release pDNAs into the cytoplasm via the “proton sponge mechanism” [17,24,32]. Therefore, PEI has been used successfully to deliver DNAs or siRNAs in GAM to animal models. For instance, a PEI complex with pDNA encoding BMP4 has shown the potential use for bone induction in a rat calvarial defect when combined with poly(lactic-co-glycolic acid) as an osteoconductive scaffold [33]. A recent study reported that cmRNA encoding BMP2 provides more bone regeneration in rat crania when applied with PEI as a complex vector [31]. However, because of its strong cationic charge, which interacts directly with negative charges on the cell surface, PEI is also known to induce cytotoxicity and agglutination. Because higher ratios of PEI to pDNA arise in higher transfection efficiencies in the clinical setting, increased cytotoxicity has a serious downside [16]. To overcome the disadvantages of cationic vectors, we previously developed a ternary complex vector by coating with anionic polymers such as polyadenylic acid, polyinosinic-polycytidylic acid, α-PGA, or γ-PGA [24]. We discovered that a complex vector coated by γ-PGA exhibited the highest gene expression, one that is comparable to pDNA/PEI complexes, without cytotoxicity and the agglutination of erythrocytes by developing a specific size and ζ-potential of this complex [24]. Biodegradable γ-PGA is capable of coating a cationic complex electrostatically to form stable anionic nanoparticles, and through a specific receptor-mediated energy-dependent process, this complex is taken in readily by cells with extremely high transgene efficiencies. In addition, to promote gene expression further, we have developed a new complex using DGL as a biodegradable cationic polymer. DGL can form a stable self-assembling nanoparticle with pDNA, and a ternary complex with this nanoparticle (pDNA-DGL-γ-PGA) displays higher gene expressions in the splenic marginal zone (rich in macrophages and dendritic cells) when intravenously injected into mice [21]. In addition, DGL-γ-PGA complexes can be taken up by caveolae-mediated endocytosis, and caveolae-mediated uptake is known not to lead to lysosomal degradation, this could be advantageous (Figure 9) [21].

Based on these findings, we focused on a ternary complex (pDNA-DGL-γ-PGA) nanoparticle (nanoball) as a nonviral vector for GAM because macrophages that first infiltrate to a bone formation site are a promising target for gene delivery. We observed the increased expression of BMP4 in cultured macrophages after treatment with BMP4-nanoballs, and this increased level was comparable in cultured MSCs and fibroblasts. In fact, transfected CD11b macrophages were detectable in GAMs with BMP4-nanoballs from the early stage of transplantation. We speculate that this may be quite advantageous for inducing bone formation because the immunoregulatory and angiogenic functions of BMP2/4 on macrophages via the pSmad1/5/8 signaling pathway and their subsequent effects on osteogenesis have been demonstrated [34]. Consistent with previous data, mRNA expression of CD206, a marker for anti-inflammatory M2 macrophages, was upregulated in transplanted GAM harboring BMP4-nanoballs, and the expression of osteocalcin mRNA subsequently increased at 2 weeks post-transplantation. Therefore, the high transfection efficiency of nanoballs in several types of cells, such as macrophages, MSCs, and fibroblasts, contributes strongly to bone induction in transplanted GAMs. Recently, miRs have been investigated for their potential usefulness in bone regeneration as a therapeutic agent because of their several advantages, including a small sequence and controlling multiple signaling pathways associated with target proteins. For instance, the injection of miR21, which regulates multiple molecular pathways in osteogenic differentiation, has been shown to promote bone fracture healing with the O-carboxymethyl chitosan matrix in rats [35]. Most recently, the delivery of tetrahedral DNA nanostructures consisting of miR335-5p, which targets corticosteroids-induced Dickkopf1 translation, with an injectable Heparin lithium hydrogel has been reported to enhance the bone regeneration by upregulating the Wnt signaling in rabbits [36]. Similarly, we have also demonstrated that miR20a in atelocollagen-based GAM could also act to augment rat calvaria bone efficiently through osteogenic and angiogenic signaling during osteogenesis [20]. In the near future, the development of GAM-based bone engineering might advance significantly by incorporating miRs into nanoballs because nanoballs must efficiently deliver multifunctional miRs to several types of localized cells.

Regarding the actual osteoinducibility of GAM in vivo, we found that GAMs harboring 1–10 μg BMP4-nanoballs markedly promoted bone augmentation in rat crania. In particular, GAM specimens with 5 μg BMP4-nanoballs induced new bone tissue by approximately threefold and 3.9-fold at 4 and 8 weeks, respectively, compared to GFP-nanoballs. Moreover, this GAM exhibited superior osteoinducibility when compared to a GAM harboring 1000 μg of naked pBMP4. These results indicate that nanoballs can deliver genes to cells infiltrating into the transplanted GAM from the surrounding host cranium tissues with high efficiency. However, to develop an ideal GAM clinically, investigating the scaffold matrixes is also essential. In this study, we applied nanoballs to a GAM matrix composed of atelocollagen and β-TCP granules to improve the low transfection efficiency associated with naked pDNAs. In preparing GAMs, nanoballs were incorporated into a 3% atelocollagen gel followed by mixing with β-TCP granules before lyophilization. Atelocollagen has frequently been examined as not only a scaffold for bone tissue engineering, but also a carrier matrix for nucleic acid delivery systems for the treatment of various diseases [37,38]. Indeed, the researchers who developed atelocollagen-based gene therapy have demonstrated its ability to deliver nucleotides, such as pDNA and antisense oligonucleotides, in vivo, which allows for the long-term gene expression [37], and also demonstrated that the optimal concentration of nucleic acids in atelocollagen for local administration is 5 μg/μL [39]. This concentration is close to our application of 1000 μg of naked pBMP4 (6 μg/μL) to GAM, even though the atelocollagen gel was lyophilized [19]. However, adapting the BMP4-nanoballs to GAM reduced the optimized concentration of pDNA in atelocollagen even further by approximately 200-fold for bone induction. Therefore, an atelocollagen-based matrix may be suitable for gradual the delivery or the release of the nanoballs into bone-injured or augmented sites because of its inherent properties in clinical applications. In addition, we applied 500–1000 μm of β-TCP granules into the GAM because these might provide an appropriate space with the proper local mechanical strength and osteoconductive properties to support the gradual release of nanoballs by atelocollagen. However, SEM analysis showed that many micro- or nano-sized β-TCP particles that separated upon mixing the granules with atelocollagen were contained in GAMs. Such smaller particles may contribute to nanoball delivery because nanoparticles of hydroxyapatite (nHA) have been shown to promote the osteocyte differentiation of rat MSCs on collagen scaffolds by acting as a nonviral vector for pBMP2 through its high affinity for DNA [40]. Thus, atelocollagen and β-TCP granules that compose a GAM matrix must support gene delivery by nanoballs efficiently and enhance in vivo osteogenesis synergistically as gene delivery agents and scaffold materials. However, a recent study reported that gelatin hydrogel is more efficient than atelocollagen as a matrix to deliver pBMP2 for bone regeneration [41]. Then, an nHA-collagen scaffold has been reported to accelerate bone regeneration with sustained localized delivery of angiogenic (pVEGF) and osteogenic (pBMP2) genes [42]. Moreover, the potential clinical usefulness of 3D printed GAM based on octacalcium phosphate (OCP) and pVEGF has been shown in the regeneration of large bone defects [43]. Therefore, many challenges in establishing GAM-based bone engineering need to be overcome by further investigating both the target genes in a nanoball and the biodegradable matrices.

## 5. Conclusions

In conclusion, GAM composed of nanoballs, which are ternary complexes of pBMP4, DGL, and γ-PGA, reliably promoted vertical bone augmentation in rats owing to high transgene efficiency. To our knowledge, this is the first report on the localized delivery of nanoballs in bone engineering. The results clearly showed that BMP4-nanoballs significantly outperformed their naked pDNA counterparts in terms of osteoinducibility. This GAM with nanoballs has immense potential in nonviral clinical gene therapy because all of the components of GAM, such as atelocollagen, β-TCP, DGL, and γ-PGA, are highly biocompatible. However, further studies are required to reveal the optimal polymer materials for a nanoball vector to deliver the osteogenic genes with proper osteoconductive and biodegradable substitutes. Meanwhile, those studies may have some limitations for widespread bone augmentation. To achieve the clinical therapy by GAM for such intractable cases, cell transplantation, such as low immunogenic immature MSCs, with GAM may be considered as a future research direction.

## Figures and Tables

**Figure 1 materials-14-07097-f001:**
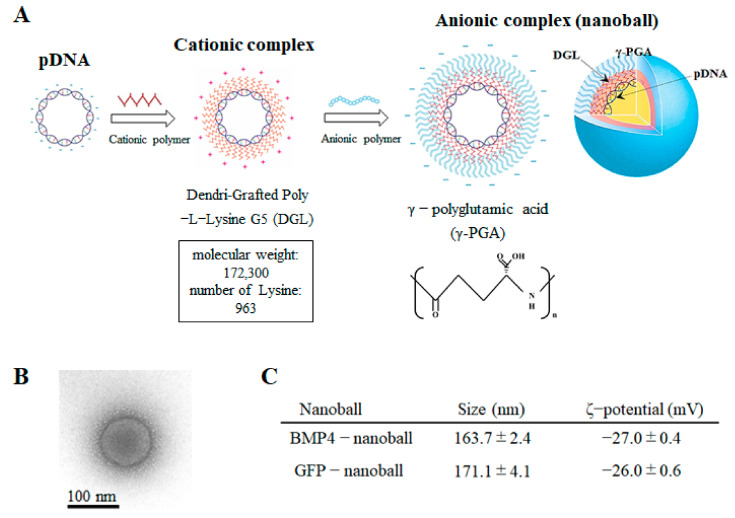
Characterization of nanoballs. (**A**) Schematic diagram presents the synthesis of nanoballs. (**B**) TEM image of a nanoball (BMP4-nanoball). There are no obvious differences between the TEM images of GFP- and BMP4-nanoballs. (**C**) Physicochemical properties of nanoballs. The data represent the mean ± SD (*n* = 6).

**Figure 2 materials-14-07097-f002:**
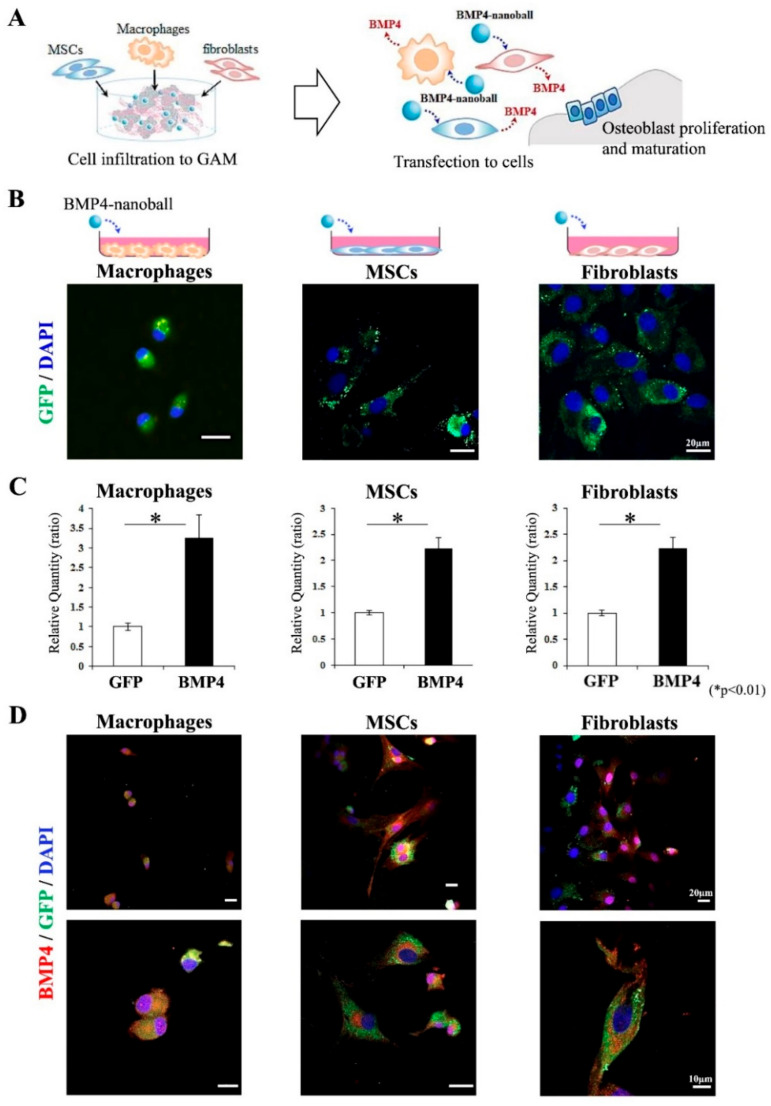
Biological activity of BMP4-nanoballs in vitro. (**A**) Schematic diagram of bone formation by infiltration of macrophages, MSCs, and fibroblasts into transplanted GAMs. (**B**) GFP expression in macrophages, MSCs, and fibroblasts at 2 h after in vitro transfection of GFP-nanoballs. Scale bar is 20 μm. (**C**) BMP4 mRNA expression in each type of cell at 48 h after transfection of GFP-nanoballs and BMP4-nanoballs (*n* = 3, * *p* < 0.01). (**D**) BMP4 production in each type of cell at 48 h post-transfection of BMP4-nanoballs. Scale bars are 20 μm (upper panels) and 10 μm (lower panels).

**Figure 3 materials-14-07097-f003:**
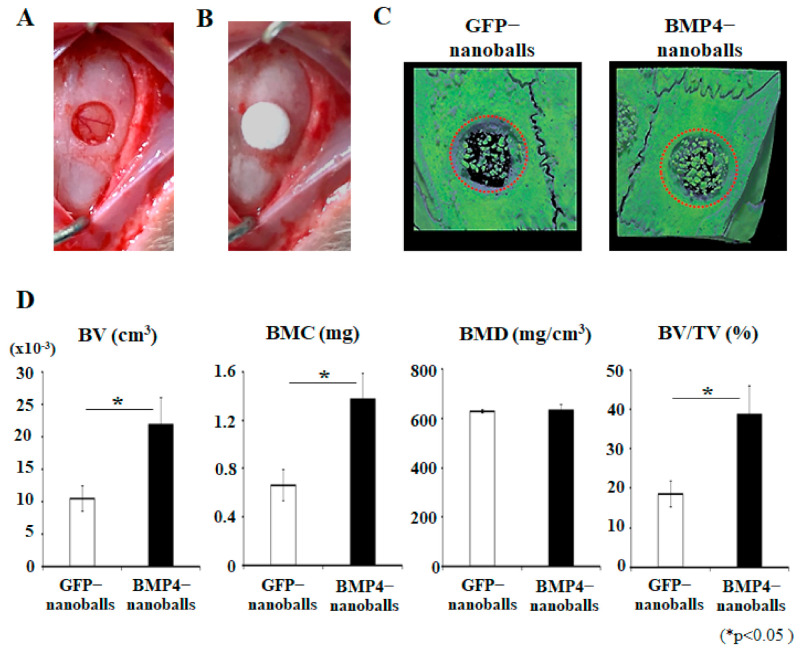
Biological function of BMP4-nanoballs in vivo. (**A**) A photograph of a created bone defect (4 mm in diameter) in rat cranium. (**B**) A photograph of BMP4-nanoballs (5 μg) transplantation with a scaffold (4 mm in diameter and 1.5 mm in thickness). (**C**) Representative reconstructed micro-CT images of specimens at 4 weeks post-transplantation. BMP4-nanoballs enhanced the bone reconstruction compared with GFP-nanoballs. Red dotted circle, original size of bone defect. (**D**) Quantification of bone reconstruction in defective sites treated with nanoballs. Bone volume (BV), bone mineral content (BMC), bone mineral density (BMD), and bone volume/tissue volume (BV/TV) were analyzed (*n* = 3, * *p* < 0.05).

**Figure 4 materials-14-07097-f004:**
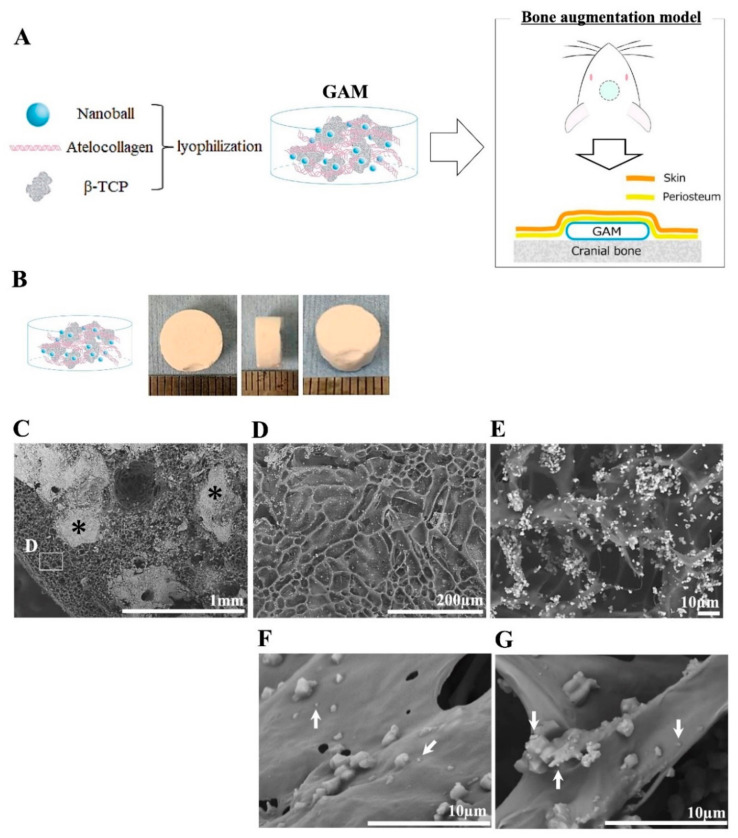
Manufactured GAM for bone augmentation. (**A**) Schematic diagram describing the experimental design for the preparation and transplantation of GAMs onto rat cranium as a bone augmented model. (**B**) Gross appearance of manufactured GAM. (**C**–**G**) SEM images of GAM composed of atelocollagen, β-TCP granules, and nanoballs. Asterisks in (**C**): 500–1000 μm β-TCP granules. Scale bar, 1 mm; (**D**) the white box area in (**C**) is shown at higher magnification. Scale bar, 200 μm; (**E**–**G**) parts of the area in (**D**) are shown at higher magnification. Scale bar, 10 μm. White arrows in (**F**,**G**): uniform nano-sized particles determined to be nanoballs at higher magnification.

**Figure 5 materials-14-07097-f005:**
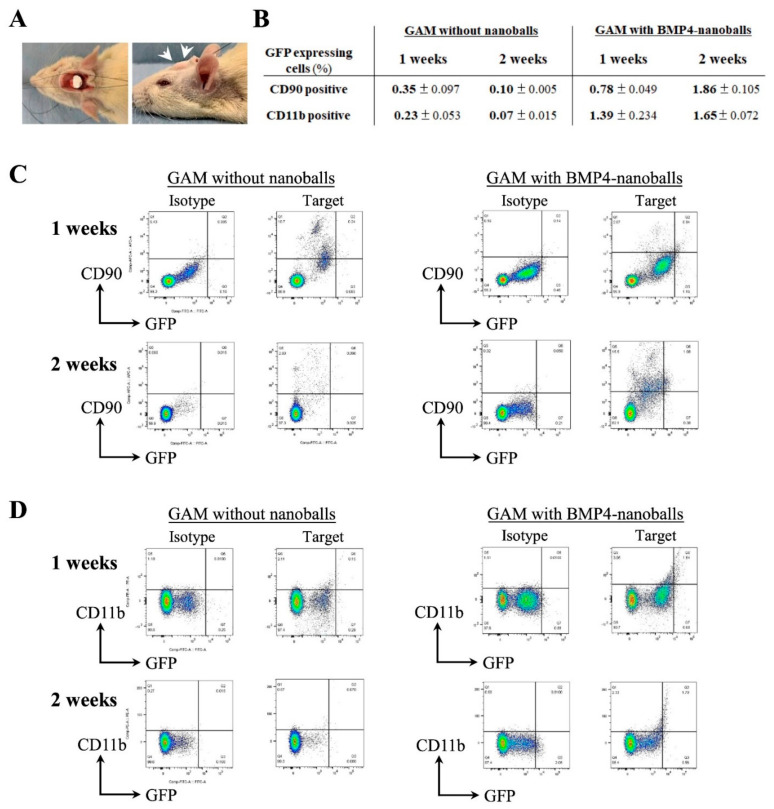
Detection of the biological activity of GAM with nanoballs (5 μg) in vivo. (**A**) Photographs of GAM transplantation onto the rat cranium bone as a bone augmentation model. White arrows indicate the GAM portion post-transplantation. (**B**) The percentages of CD90- (as an MSC marker) or CD11b (as a macrophage marker)-positive cells in GFP-expressing transfected cells isolated from GAM specimens at 1- or 2-weeks post-transplantation. The data represent the mean ± SD (*n* = 5). (**C**) Density plots from flow cytometry for CD90- and GFP-positive subpopulations in GAM with/without BMP4-nanoballs at 1- or 2-weeks post-transplantation. (**D**) Density plots from flow cytometry for CD11b- and GFP-positive subpopulations in GAMs with/without BMP4-nanoballs at 1- or 2-weeks post-transplantation.

**Figure 6 materials-14-07097-f006:**
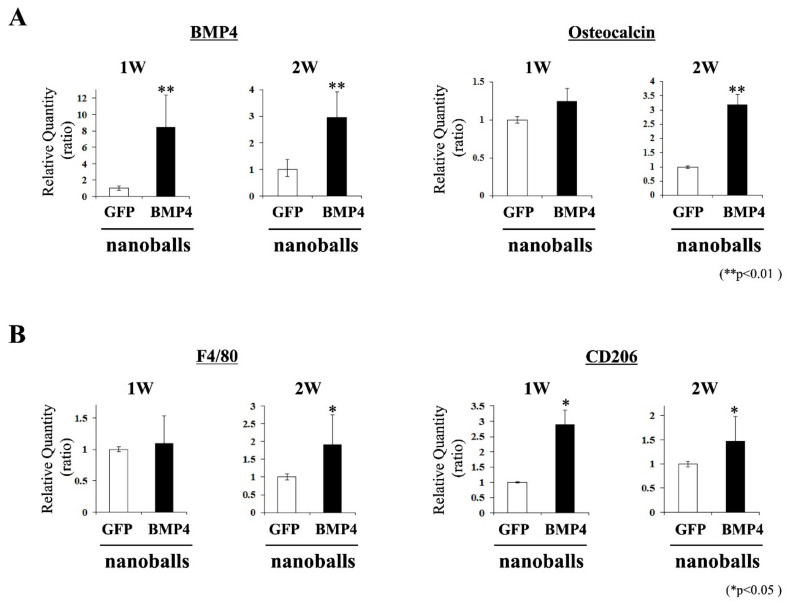
Expression of osteoblast- and macrophage-differentiation genes. (**A**) mRNA expressions of BMP4 and osteocalcin (as markers of osteoblastic differentiation). (**B**) mRNA expressions of F4/80 and CD206 (as markers of M2 macrophages) (*n* = 6, * *p* < 0.05, ** *p* < 0.01).

**Figure 7 materials-14-07097-f007:**
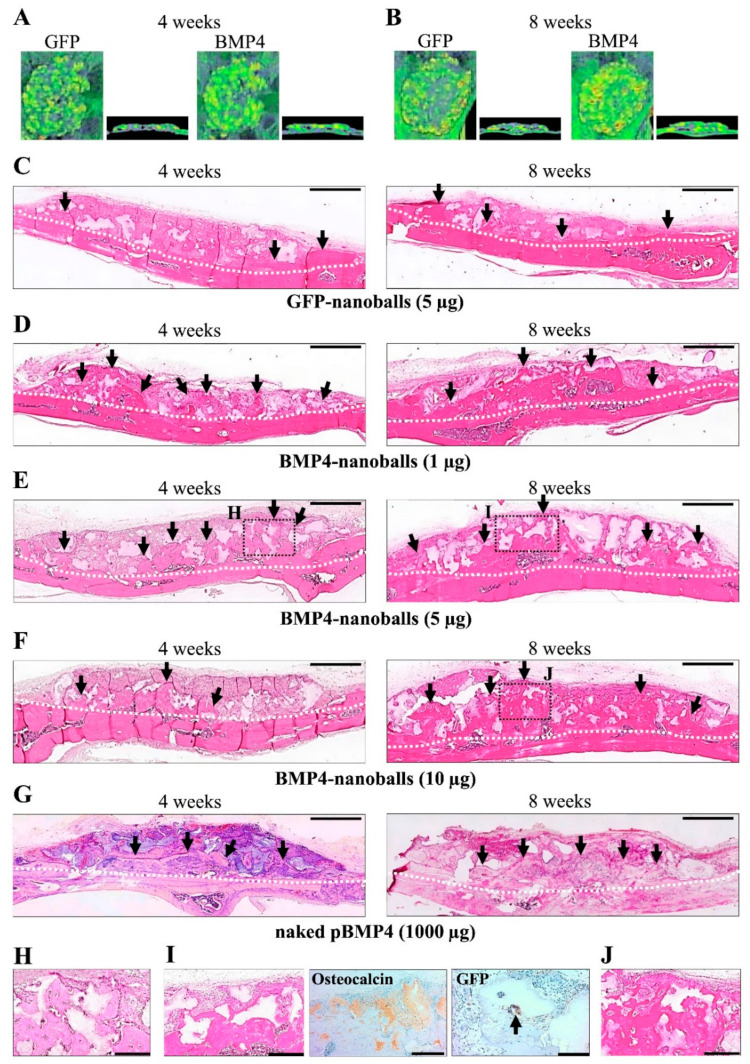
Micro-CT and histological appearances of augmented bone at 4 and 8 weeks after GAM transplantation (*n* = 5 per group, respectively). (**A**,**B**) Representative images (axial and sagittal views) of micro-CT in specimens of GAM with GFP-nanoballs and BMP4-nanoballs. (**C**–**G**) Representative images (H&E staining) of augmented bone tissues in GAM specimens with 5 μg of GFP-nanoballs (**C**), 1 μg of BMP4-nanoballs (**D**), 5 μg of BMP4-nanoballs (**E**), 10 μg of BMP4-nanoballs (**F**), or 1000 μg of naked pBMP4 (**G**). Scale bar, 1 mm; white dotted line, boundary of the cranium and newly formed bone; black arrow, augmented bone area. (**H**–**J**) The black box areas in (**E**,**F**) are shown at higher magnifications. (**I**) Osteocalcin and GFP expressions in a specimen of (**E**) at 8 weeks. Scale bar is 200 μm.

**Figure 8 materials-14-07097-f008:**
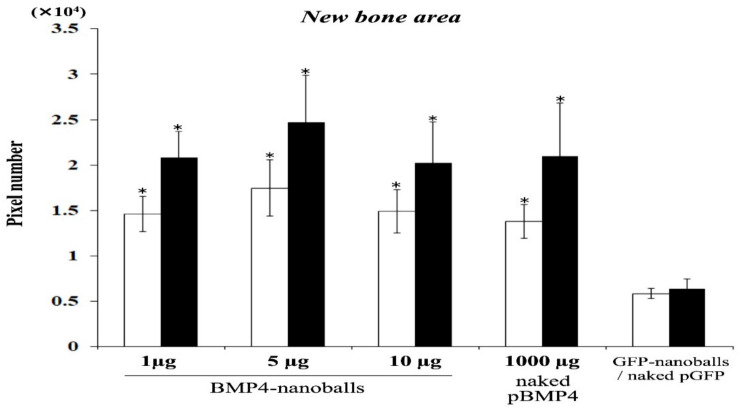
Quantification of augmented bone area at 4- and 8-weeks post-transplantation (*n* = 25 per group, * *p* < 0.01).

**Figure 9 materials-14-07097-f009:**
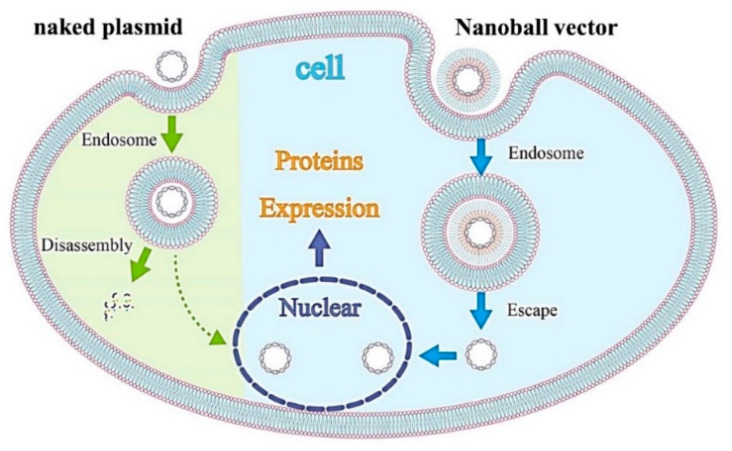
Schematic diagram of the transfection mechanism by a nanoball. Anionic materials have difficulty passing through the cell membrane by simple diffusion because of electrostatic repulsion by the membrane. Therefore, they are considered to be taken up into cells via caveolae-mediated endocytosis after binding with a specific receptor on the cell surface. After being taken up, they are generally disassembled in late endosomes or lysosomes. For nanoballs, endosome escape is easy because of the proton sponge effect by the high three-dimensional spatial ability of DGL. After escaping from the endosome and being released into the cytoplasm, the protein is expressed by the translocation of the plasmid to the nucleus. However, the mechanistic details of intracellular transport from endosomes to the nucleus remain elusive.

**Table 1 materials-14-07097-t001:** Rat primer sets.

Gene	Forward Primer	Reverse Primer
*bmp4*	5′-CACTGTGAGGAGTTTCCATCAC-3′	5′-AGGAGATCACCTCATTCTCTGG-3′
*f4/80*	5′-ACCTGCCACAACACTCTTGG-3′	5′-TCACAAGATTCTTCCAGGCCC-3′
*cd206*	5′-TTCCTTTGGACAGACGGACG-3′	5′-TCCCTGCCTCTCGTGAATTG-3′
*gapdh*	5′-TGCACCACCAACTGCTTAG-3′	5′-GGATGCAGGGATGATGTTC-3′

**Table 2 materials-14-07097-t002:** Antibodies for immunostaining and flow cytometry.

Antibodies	Company, Catalog No.
Rabbit polyclonal anti-BMP4 antibody	Abcam, ab39973
Rabbit polyclonal anti-Osteocalcin antibody	Abcam, ab93876
Rabbit polyclonal anti-GFP antibody	Abcam, ab6556
Goat anti-Rabbit IgG (H + L) Highly Cross-Adsorbed Secondary Antibody, Alexa Fluor 546	Invitrogen, A-11035
Mounting Medium with DAPI	Vector Laboratories, H-1200
APC anti-rat CD90/mouse CD90.1 (Thy-1.1) Antibody	Biolegend, 202526
PE anti-rat CD11b/c Antibody	Biolegend, 201807
APC Mouse IgG1, κ Isotype Ctrl Antibody	Biolegend, 400119
PE Mouse IgG2a, κ Isotype Ctrl Antibody	Biolegend, 400211

## Data Availability

The data presented in this study are available on request from the author.

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
