# Peer review of "Gene-Activated Matrix with Self-Assembly Anionic Nano-Device Containing Plasmid DNAs for Rat Cranial Bone Augmentation"

_materials, 2021, doi:10.3390/ma14227097_

Round 1

Reviewer 1 Report

This manuscript entitled “Gene-activated matrix with self-assembly anionic nano-device containing plasmid DNAs for rat cranial bone augmentation” by Masahito Hara et al. reports an  anionic vector to appropriate matrices can facilitate GAM-based bone engineering. However, I have some questions to be elucidated.

  1. Which nanoball of TEM image in figure 1B refer to?GFP-nanoballs or BMP4-nanoballs? The GFP should be a separate test.
  2. We known that gene transfection should proceed with some medium, such as Lipo2000. What is the mechanisms of the nanoball?
  3. How do the authors confirm the plasmid DNAs in the nanoballs? Please supply related characterization data.
  4. Figure 2C and Figure 6 must be verified by Western Blot.
  5. Add some latest references.

Reviewer 2 Report

I have reviewed the manuscript “Gene-activated matrix with self-assembly anionic nano-device containing plasmid DNAs for rat cranial bone augmentation” submitted to “Materials” for publication. In this study, authors have developed nanoballs, a biocompatible self-assembly nano-vector based on electrostatic interactions arranged anionic macromolecules to polymeric nanomaterials to create nucleic acid carriers. This manuscript fits well within the scope of the journal; it needs some improvements; there are a few suggestions that authors may consider to improve it further:

The use of English language is reasonable, however, there are a number of punctuation and grammatical errors; that should be corrected and rephrased using academic English for a better flow of text for reader.

Authors should make sure to define all the abbreviations at their first appearance in the text.

- Abstract is appropriate, however the statement of objective is not clear, please revise the objective of the study.

Line 30: please delete (198 words)

Line 39: please cite this statement with a reference

- The introduction covers all the background information however seems a bit lengthy with mild repetition of information. But acceptable.

Methods, data, and results are very comprehensively and precisely presented.

Figures 3 and 4 cannot be ahead of figure 2. Please re-order.

Figure 9 in the discussion: although it is a worthy figure, it is very rare to using figures in the discussion section. Particularly when there is no citations suggesting this is authors’ own image and there is no copyright issues?

I would suggest presenting conclusions as a separate distinct section after discussion.

- Please write Limitations and the future research directions in the discussion section.

Round 2

Reviewer 1 Report

The author solved my problems and I agree to publish.

Reviewer 2 Report

The revised manuscript has been improved and addressed all the comments. thank you